# Relationship Between Alcohol Consumption and Vascular Structure and Arterial Stiffness in Adults Diagnosed with Persistent COVID: BioICOPER Study

**DOI:** 10.3390/nu17040703

**Published:** 2025-02-16

**Authors:** Silvia Arroyo-Romero, Leticia Gómez-Sánchez, Nuria Suárez-Moreno, Alicia Navarro-Cáceres, Andrea Domínguez-Martín, Cristina Lugones-Sánchez, Olaya Tamayo-Morales, Susana González-Sánchez, Ana B. Castro-Rivero, Marta Gómez-Sánchez, Emiliano Rodríguez-Sánchez, Luis García-Ortiz, Elena Navarro-Matías, Manuel A. Gómez-Marcos

**Affiliations:** 1Primary Care Research Unit of Salamanca (APISAL), Salamanca Primary Care Management, Institute of Biomedical Research of Salamanca (IBSAL), 37005 Salamanca, Spain; silvia_ar@usal.es (S.A.-R.); nuria.suarez@usal.es (N.S.-M.); alicia.nav@usal.es (A.N.-C.); andreadm@usal.es (A.D.-M.); crislugsa@gmail.com (C.L.-S.); olayatm@usal.es (O.T.-M.); gongar04@gmail.com (S.G.-S.); emiliano@usal.es (E.R.-S.); lgarciao@usal.es (L.G.-O.); enavarro@saludcastillayleon.es (E.N.-M.); 2Castilla and León Health Service-SACYL, Regional Health Management, 37005 Salamanca, Spain; 3Emergency Service, University Hospital of La Paz P. of Castellana, 261, 28046 Madrid, Spain; letici.gomez@salud.madrid.org; 4Research Network on Chronicity, Primary Care and Health Promotion (RICAPPS), 37005 Salamanca, Spain; 5Home Hospitalization Service, Marqués of Valdecilla University Hospital, s/n, 39008 Santander, Spain; martagmzsnchz@gmail.com; 6Department of Medicine, University of Salamanca, 28046 Salamanca, Spain; 7Department of Biomedical and Diagnostic Sciences, University of Salamanca, 37007 Salamanca, Spain

**Keywords:** alcohol consumption, vascular structure, vascular function, pulse wave velocity, intima–media thickness, cardiovascular index, long COVID

## Abstract

Background: The relationship between alcohol consumption and vascular structure and arterial stiffness is not clear, especially in people diagnosed with persistent COVID. The aim of this study was to evaluate how alcohol use is related to vascular structure and arterial stiffness in adults with persistent COVID. Methods: A descriptive cross-sectional study was conducted involving 305 individuals (97 men and 208 women) diagnosed with persistent COVID according to the WHO criteria. Arterial stiffness was assessed by measuring the cardio-ankle vascular index (CAVI) and the brachial-ankle pulse wave velocity (ba-PWV) with a VaSera VS-1500 device, and the carotid-femoral pulse wave velocity (cf-PWV) with a Sphygmocor device. Vascular structure was assessed by measuring carotid intima-media thickness (c-IMT) with a Sonosite Micromax ultrasound unit. Alcohol intake was calculated using a standardized questionnaire and quantified in g/week. Results: Mean alcohol intake was 29 ± 53 g/week (men 60 ± 76 g/w and women 15 ± 27 g/w; *p* < 0.001). Heavy drinkers showed higher levels of c-IMT, cf-PWV, ba-PWV and CAVI than non-drinkers (*p* < 0.05). The multinomial regression analysis adjusted for sex and lifestyles showed a positive association between heavy drinking and c-IMT and cf-PWV values (β = 1.08 (95% CI 1.01–1.17); β = 1.37 (95% CI 1.04–1.80); ba-PWV and CAVI figures showed a similar trend, without reaching statistical significance. Conclusions: The results of this study indicate that high alcohol use in patients with persistent COVID is linked to higher c-IMT and cf-PWV figures than in non-drinkers.

## 1. Introduction

Alcohol use is a global health issue and one of the main modifiable risk factors [1]. A J-shaped association has been described between alcohol use and cardiovascular disease [2,3], with low consumption being linked to a lower incidence of ischemic heart disease [4] and increased levels of high-density lipoprotein cholesterol (c-HDL) [5], while excessive drinking raises cardiovascular risk [6]. Some studies, however, indicate that drinking any amount of alcohol causes increased blood pressure [7] and body mass index (BMI) [8].

Alterations of vascular structure and arterial stiffness are considered predictors of cardiovascular risk [9]. The alteration of vascular structure can be assessed non-invasively by performing a carotid ultrasound, allowing the carotid intima–media thickness (c-IMT) to be determined [10]. C-IMT is an early marker of atherosclerosis [11], and an increase is associated with a higher risk of ischemic heart disease [12,13] and cerebrovascular accidents [14,15]. Another sensitive marker of cardiovascular risk is arterial stiffness [16,17], which can be measured with carotid-femoral pulse wave velocity (cf-PWV), brachial-ankle pulse wave velocity (ba-PWV), and the cardio-ankle vascular index (CAVI) [18]. Arterial stiffness is a predictor of cardiovascular events and all-cause morbidity and mortality [19,20,21]. It has also been linked to a higher risk of arterial hypertension (AHT) [22], diabetes mellitus (DM) [23] and kidney diseases [24], among others. Alcohol use also plays a key role in arterial stiffness and vascular damage [25] since excessive drinking is associated with increased arterial stiffness [2,26,27,28] and increased c-IMT [29].

Meanwhile, infection by SARS-CoV 2 triggers an inflammatory response that has been linked to cardiovascular complications [30]. Several studies have shown a significant increase in cf-PWV [31] and CAVI [32] after an acute SARS-CoV 2 infection. At present, understanding persistent COVID is becoming more important given its predominance over acute infection. Thus, increased risks of developing AHT [33], DM [34] and dyslipidemia [35] have been observed in patients with persistent COVID in a bidirectional relationship [36,37] and greater arterial stiffness has been described in patients with persistent COVID compared to healthy controls [38], linked to the time elapsed since the onset of the disease [39]. However, the mechanisms underlying this increased cardiovascular risk and the perpetuation of symptoms in patients with persistent COVID are as yet not well established. The influence of lifestyle habits, such as smoking [40] or alcohol use [41], has been posited, while alcohol abuse has been observed in patients with previous SARS-CoV 2 infections [42] and in particular in patients with persistent COVID [43]. This pattern of consumption could contribute to increased arterial stiffness. In this context, it is essential to study the influence of alcohol use on the vascular health of patients with persistent COVID. The main objective of this study was, therefore, to analyze the relationship of alcohol use with vascular structure and arterial stiffness in adults diagnosed with persistent COVID.

## 2. Materials and Methods

### 2.1. Study Design

The results presented in this study are part of the BioICOPER study, the protocol of which has been published previously [44]. The BioICOPER study is a descriptive cross-sectional study, carried out at the Salamanca Primary Care Research Unit (APISAL) and registered in ClinicalTrials.gov with registration number NCT05819840 in April 2023.

### 2.2. Study Population

Using consecutive sampling, following the clinical history records of primary care and persistent COVID consultations in Salamanca’s internal medicine service, 305 participants with a persistent COVID diagnosis were recruited. The diagnostic criteria defined by the World Health Organization (WHO) were used [45], i.e., patients who, within three months after SARS-CoV-2 infection, and without any other apparent cause, present symptoms lasting at least two further months. These symptoms may persist from the original infection or appear anew after an initial improvement, and may fluctuate over the course of the disease [45]. Subjects excluded were those in a terminal state, unable to travel to the health center, with a history of cardiovascular disease (ischemic heart disease or cerebrovascular event) or with a glomerular filtration rate under 30 mL/min/1.73 m^2^.

### 2.3. Variables and Measurement Instruments

Four healthcare professionals, previously trained following a standardized protocol, performed the necessary examinations and administered the questionnaires collected during the study. A researcher external to the study was responsible for data quality control.

#### 2.3.1. Sociodemographic Variables and Personal History

On inclusion in the study, the participant’s age, sex, and personal history of hypertension, dyslipidemia, DM, and use of medicines were recorded. The date of acute SARS-CoV-2 infection diagnosis was also noted in order to calculate the duration of the disease (up to the date of assessment).

#### 2.3.2. Lifestyles

##### Alcohol Consumption

Alcohol intake was determined by a structured questionnaire, recording the amount and type of alcoholic beverages drunk during the previous 7 days in grams/week. Participants were then distributed into the following groups according to the Spanish Ministry of Health criteria: ‘non-drinker’, ‘low consumption’, ‘moderate consumption’ and ‘heavy consumption’ [46]. For statistical analysis, the ‘low consumption’ and ‘moderate consumption’ groups were combined into a single ‘low-moderate consumption’ group.

##### Adherence to the Mediterranean Diet

We assessed adherence to the Mediterranean diet (MD) using the 14-question Mediterranean Diet Adherence Screener (MEDAS) questionnaire, developed by the PREDIMED group and previously validated in the Spanish population [47].

##### Tobacco Consumption

To assess smoking habits, we used a standard four-question questionnaire adapted from the WHO MONICA project [48], recording years of smoking.

##### Physical Activity

Physical activity was measured with the WHO Global Physical Activity Questionnaire (GPAQ) [49]. This questionnaire comprises 16 questions, assesses physical activity at work, commuting and leisure time over the past week, and provides the result in metabolic equivalent of task per minute per week (METs-min/week).

#### 2.3.3. Cardiovascular Risk Factors

##### Measurement of Blood Pressure (BP)

To measure systolic BP (SBP) and diastolic BP (DBP), three measurements were taken with the OMRON sphygmomanometer model M10-IT (Omron Healthcare, Kyoto, Japan), recording the average of the last two. Following the recommendations of the European Society of Hypertension (ESH) [50], measurements were taken on the dominant arm of the seated participant after they had rested for at least five minutes. Pulse pressure (PP) was calculated using the formula PP = SBP − DBP, and mean arterial pressure (MAP) was calculated using the formula MAP = (2DBP + SBP)/3. The OMRON blood pressure monitor also provided the heart rate (HR) value. Hypertension was diagnosed in participants taking antihypertensive drugs or with BP values ≥140/90 mmHg.

##### Analytical Tests

Analytical parameters were determined by collecting venous blood samples between 08:00 and 09:00 am, with participants fasting, not smoking nor drinking alcohol or caffeine in the previous 12 h. The following analytical parameters were assessed: fasting plasma glucose (FPG), HbA1c, total cholesterol, low-density lipoprotein (LDL) cholesterol, HDL cholesterol and triglycerides. Participants were considered to have DM if they were taking hypoglycemic drugs or if they had FPC levels ≥ 126 mg/dL or HbA1c ≥ 6.5%. Dyslipidemia was diagnosed if lipid-lowering drugs were taken or with values of total cholesterol ≥240 mg/dL, LDL-C ≥ 160 mg/dL, HDL-C < 40 mg/dL in men and <50 mg/dL in women, or triglycerides ≥ 150 mg/dL.

##### Anthropometric Measurements

Weight (kg) was measured using the InBody 230 monitor (InBody Co., Ltd., Seoul, Republic of Korea), with the patient fasting for 2 h and wearing light clothing without shoes. Height was measured with the patient barefoot, using a stadiometer (Seca 222, Medical Scale and Measurement Systems, Birmingham, UK). Body mass index (BMI) was calculated by dividing weight (kg) by height squared (m^2^), with BMI ≥ 30 kg/m^2^ considered as obesity. Waist circumference was determined with a tape measure above the iliac crests, with the patient exhaling, standing and in underwear, following the recommendations of the Spanish Society for the Study of Obesity [51]. A circumference of ≥102 cm in men and ≥88 cm in women was classified as abdominal obesity.

#### 2.3.4. Vascular Structure and Arterial Stiffness

##### Vascular Structure

Vascular structure was assessed by carotid wall intima–media thickness (c-IMT). C-IMT was measured using the Sonosite Micromax ultrasound system (FUJIFILM Sonosite Washington, DC, USA), with a high-resolution linear probe and Sonocal software series 3000 performing automatic measurements based on a protocol [15]. Ultrasound imaging was performed by previously trained personnel. A 10 mm portion of the common carotid artery was selected, located 1 cm from the bifurcation. Measurements were made of the proximal and distal wall in lateral, anterior and posterior projections. Measurements were obtained with the participant supine, head extended and tilted towards the contralateral shoulder [15].

##### Arterial Stiffness

Arterial stiffness was analyzed by measuring carotid-femoral pulse wave velocity (cf-PWV), brachial-ankle pulse wave velocity (ba-PWV), and cardio-ankle vascular index (CAVI) [16].

For the assessment of cf-PWV, the SphygmoCor device (AtCor Medical Pty Ltd., head office, West Ryde, Australia) was used with the patient lying down. Time was estimated by analyzing the delay of the pulse waves of the carotid, radial and femoral arteries with respect to the R wave of the electrocardiogram (ECG). Distance was determined using a tape measure from the sternal jugulum to the sensor location on the carotid and radial or carotid and femoral arteries [52].

CAVI and ba-PWV were assessed with a VaSera VS-2000 device (Fukuda Denshi Co., Ltd., Tokyo, Japan). Electrodes were attached to the arms and ankles of the silent and motionless participant, and a heart sound microphone was secured in the second intercostal space. CAVI was calculated using the following equation: stiffness parameter β = 2ρ × 1/(SBP − DBP) × ln(SBP/DBP) × PWV, where ρ is blood density and PWV is measured between the aortic valve and the ankle. Measurements were assumed valid after three consecutive heartbeats [53]. Ba-PWV was estimated with the equation: ba-PWV = (0.5934 × height in cm + 14.4724)/tba, where tba is the time interval between the brachial and ankle waveform [54].

### 2.4. Statistical Analysis

First, we used the Kolmogorov–Smirnov test to check the distribution of variables. The comparison of means for independent samples with two categories was performed with the Student’s *t* test or with the Mann–Whitney U test, depending on whether the distribution of the variables was normal or not. The comparison of two proportions was made with the chi-square test, while the comparison of means for independent samples with more than two categories was performed with the ANOVA test or with the Kruskal–Wallis H test, depending on whether not or the variables were normally distributed. The post hoc tests were conducted with the DMS test. The comparison of means adjusted for age and sex of the vascular structure and function measures according to alcohol use was performed with the UNIANOVA test. To analyze the correlation between alcohol use and the vascular structure and function variables, we used the Spearman Rho coefficient. The association between alcohol use and the parameters of vascular structure and function was performed using several multinomial regression analysis models. The dependent variable was alcohol use, classified into three categories and coded as: non-drinker = 0 (used as reference), low-moderate consumption = 1 and heavy consumption = 2. The independent variables were c-IMT, in mm, cf-PWV and ba-PWV, in m/s, and CAVI. The adjustment variables were age in years and different lifestyles (MD score, years of smoking and total physical activity measured in METs/min/week). Several analyses were performed globally and by sex. The statistical program SPSS for Windows, v28.0 (IBM Corp, Armonk, NY, USA) was used for the analyses, and the cut-off point for statistical significance was *p* < 0.05.

### 2.5. Ethical Principles

The study was approved on 27 June 2022 by the “Medicine Research Ethics Committee of the Salamanca Health Area” (CEIm reference code: Ref. PI 2022 06 1048). The recommendations of the Declaration of Helsinki [55] and the WHO were respected during the study. Confidentiality was guaranteed in accordance with Organic Law 3/2018, European Regulation 2016/679 and European Council 27 April 2016 on data protection. All participants signed the informed consent before being included in the study and after being informed of the examinations and questionnaires involved.

## 3. Results

### 3.1. Participant Characteristics

The characteristics of the subjects included in this study overall and by sex are shown in Table 1. More women were included in the study than men (208 vs. 97). Men consumed more alcohol, had higher blood pressure, blood glucose, triglycerides, BMI, waist circumference, and vascular structure and function parameters than women. Women presented higher HDL cholesterol levels than men. The time elapsed from the diagnosis of acute SARS-CoV 2 infection to inclusion in the study was 39 ± 9.6 months.

Subject characteristics stratified by alcohol use (g/week) into three categories (non-drinker, low-moderate consumption and heavy consumption) are shown in Table 2. ‘Heavy consumption’ presented higher mean values in the vascular structure and function parameters than ‘non-drinker’; c-IMT (*p* = 0.002), cf-PWV (*p* = 0.007), ba-PWV (*p* = 0.03) and CAVI (*p* = 0.008). The results by sex are shown in men in Appendix A and in women in Appendix A.

### 3.2. Vascular Structure and Arterial Stiffness by Tertiles of Alcohol Consumption

Figure 1 shows a box and whisker plot of vascular structure and function measures by level of alcohol use. The non-drinker group had lower mean values c-IMT and cf-PWV than the heavy consumption group (*p* = 0.008 and *p* = 0.02), while ba-PWV and CAVI showed lower mean values in the non-drinker group and in the low-moderate consumption group than in the heavy consumption group (ba-PWV: *p* = 0.009 y *p* = 0.02; CAVI: *p* = 0.003 y *p* = 0.02).

The age- and sex-adjusted marginal means of vascular structure and function measures by amount of alcohol consumed are shown in Figure 2. The low-moderate consumption group had lower vascular function values than the non-drinker or heavy consumption group, although this was only significant for the ba-PWV figures (*p* = 0.002) and not c-IMT (*p* = 0.38), cf-PWW (*p* = 0.10) and CAVI (*p* = 0.19).

### 3.3. Vascular Structure and Arterial Stiffness According to Type of Drink

The mean values of vascular structure and function measures by beverage type overall are shown in Table 3. Wine drinkers showed higher values of all vascular structure and function measures analyzed than non-drinkers, with statistical significance in c-IMT (*p* = 0.02) and CAVI (*p* = 0.03). Beer drinkers had higher values of all vascular structure and function measures analyzed than non-drinkers, but the difference was only significant in c-IMT (*p* = 0.01). Drinkers of high alcohol content beverages showed higher values in all vascular structure and function measures analyzed than non-drinkers.

Figure 3 shows the percentage of drinkers of wine, beer or high-alcohol beverages overall and by sex. The percentage of men was higher than that of women in the group for high-alcohol drinks (*p* < 0.001).

### 3.4. Association Between Alcohol Use and Vascular Structure and Function

Alcohol use in grams/week was found to have a positive correlation with c-IMT (Rho = 0.15, *p* = 0.008), cf-PWV (Rho = 0.17, *p* = 0.12), ba-PWV (Rho = 0.08, *p* = 0.06) and CAVI (Rho = 0.14, *p* = 0.02).

The results of the multinomial regression analysis are shown in Table 4 and Figure 4. For each 10-micrometer increase in c-IMT, the β of belonging to the heavy consumption group (vs. non-drinker) was β = 1.08 (95% CI: 1.01–1.17). For each 1 cm increase in cf-PWV, the β of belonging to the heavy consumption group (vs. non-drinker) was β = 1.37 (95% CI: 1.04–1.80). For each 1 cm increase in ba-PWV, the β of belonging to the heavy consumption group (vs. non-drinker) was β = 1.36 (95% CI: 0.94–1.95). For each 1-unit increase in CAVI, the β of belonging to the heavy consumption group (vs. non-drinker) was β = 1.68 (95% CI: 0.79–3.58), all results after adjusting for age and lifestyle.

## 4. Discussion

This study provides novel results regarding the relationship between alcohol use and data on the type of alcohol and vascular structure (measured by c-IMT) and arterial stiffness (assessed by cf-PWV, ba-PWV and CAVI) in a sample of patients diagnosed with persistent COVID. The main findings suggest that heavy drinking is associated with worse c-IMT and cf-PWV values than in non-drinkers. The ba-PWV and CAVI figures revealed a similar although not significant trend.

### 4.1. Vascular Structure and Arterial Stiffness in Patients with Persistent COVID

The mean values of arterial structure and stiffness parameters in our participants were similar to those published in patients with persistent COVID in Texas [56] and Africa [38], with mean cf-PWV values of 7.3 m/s and 7.1 m/s, respectively. However, only women were analyzed in the first study [56], and higher cf-PWV values were observed in women with persistent COVID than in those who were healthy (7.1 m/s vs. 6.0 m/s). A similar comparison can be made with a study carried out in the population of Salamanca involving 501 healthy subjects selected by random sampling stratified by age group and sex, in the EVA study [57]. This showed lower values in population groups of the same age as in the present study; thus, in the case of women with this mean age the values were cf-PWV: 5.7 m/s; ba-PWV: 11.7 m/s and CAVI: 7. The comparisons in men were similar.

### 4.2. Alcohol Consumption and Vascular Structure (c-IMT)

Several studies have explored the relationship between alcohol use and c-IMT. The results obtained in the present study are consistent with those of previous studies carried out in the population without persistent COVID. The EVA study [26] described higher levels of c-IMT in subjects with high alcohol consumption (0.73 ± 0.1 mm) compared to non-drinkers (0.67 ± 0.1 mm, *p* < 0.05) or moderate consumers (0.64 ± 0.0 mm, *p* < 0.05), a relationship which had been described previously in a prospective cohort study carried out in young adults (non-drinkers: 0.57 ± 0.0 mm, low-moderate consumers 0.59 ± 0.0 mm, high alcohol consumption 0.6 ± 0.0 mm) [58]. Similarly, the STRATEGY [59] and MASALA studies [60] found a relationship between high alcohol intake and increased c-IMT, independent of other cardiovascular risk factors (c-IMT increases by 0.025 mm per 21.4 g/day intake of alcohol [59] and by 0.096 mm in >7 drinks/week consumption [60]. These findings suggest that drinking alcohol could have proatherogenic effects. Other studies have postulated a J-shaped association between alcohol use and c-IMT, with moderate consumption groups showing lower c-IMT levels than non-drinkers (moderate consumption: 0.77 mm vs. non-drinkers: 0.78 mm) [61]. Such lower c-IMT values compared to non-drinkers may result from the anti-inflammatory effect of low alcohol consumption, with a reduction in the atherosclerotic load mediated by polyphenols [62]. Nevertheless, moderate consumption stimulates the hepatic secretion of apolipoproteins and lipoproteins, and increases triglyceride concentrations [63]. In contrast, some studies have shown an inverse relationship between drinking alcohol and c-IMT [64,65,66]. In the study by Kim et al. [66] this association disappeared after adjusting for dyslipidemia. Moreover, the study excluded subjects being treated with antihypertensive, hypoglycemic or lipid-lowering drugs, so the results cannot be extrapolated to the Spanish population, where drugs to control cardiovascular risk factors are widely used [67], and do not cover patients with high cardiovascular risk. Conversely, the study by Laguzzi et al. [65] only included patients with high cardiovascular risk. The differences between the Laguzzi study and our research may be due to the fact that alcohol use attenuates the relationship between saturated fats and c-IMT in patients with high cardiovascular risk [68].

### 4.3. Alcohol Consumption and Arterial Stiffness (cf-PWV, ba-PWV and CAVI)

Several studies conducted in the population without persistent COVID have found a J-shaped relationship between alcohol use and arterial stiffness measured by cf-PWV. Thus, the EVA study revealed a positive association (β = 0.02 (0.01–0.37)) with a J pattern (cf-PWV in non-drinkers: 6.3 m/s, in moderate drinkers: 5.8 m/s and in heavy drinkers: 6.5 m/s) [26]. The present study also showed a lower cf-PWV value in low-moderate drinking compared to non- or excessive drinking in the estimation of marginal means. It has been suggested that this decrease in arterial stiffness in low-moderate drinkers compared to abstainers is due to the fact that the abstainer group included patients who stopped drinking for health reasons and thus had a higher cardiovascular risk [69]. However, in a prospective study which separated ex-drinkers from abstainers, the lower cf-PWV figures in the moderate consumption group was maintained in comparison to the abstainer group (8.3 m/s vs. 8.8 m/s) [70]. In this way, the Sierksma study used the consumption of 0–3 drinks/week as reference and the J pattern was maintained: lower cf-PWV levels in moderate alcohol consumption (4–10 drinks/week) compared to non-drinkers or those consuming small amounts (β = −0.77 m/s (−1.26 to −0.28) [71]). The J pattern suggests that low alcohol consumption could exert a protective effect, mediated by improvements in the lipid profile (increased HDL-c) and endothelial function (nitric oxide production and decreased inflammatory mediators) [72,73]). Conversely, heavy drinking has been linked to the proliferation of vascular smooth muscle and fibroblasts [74], and the production of collagen and matrix metalloproteinase (which degrades the elastin of the vascular wall) [75]. These changes are mediated by norepinephrine [74] and lead to an increase in arterial stiffness [76]. On the other hand, some studies analyzing subjects with low-moderate (not high) consumption have failed to find a significant association between alcohol use and cf-PWV [77]. Other reviews, based on mendelian randomization studies [69], have also found no association between alcohol consumption and arterial stiffness, possibly due to the removal of confounding epigenetic factors (inherent to humans). However, the observational analysis of these studies describes a U-shaped association [69].

Regarding ba-PWV, and in line with our results, the EVA study yielded a positive relationship that was not statistically significant (β = 0.05 (−0.37 to 0.48)) [26]. A recent study in the Japanese population described a progressive increase in ba-PWV as a function of the alcohol dose (β = 1.84, *p* < 0.05), especially in participants aged under 50 years [78], while other studies have described this association only in men (men *p*-value < 0.036, women *p*-value = 0.056) [79]. The lack of statistical significance in women can be explained by the low number of women in the group of excessive drinkers. On the other hand, the relationship with CAVI has not been extensively studied. As with ba-PWV, both this study and the EVA study showed a non-significant positive association: β = 0.17 (−0.71 to 0.41) [26].

Other studies analyzing arterial stiffness by photoplethysmography have also found a positive association between alcohol use and higher levels of the arterial stiffness index in both sexes [80].

Regarding acute alcohol intake, this is linked to better arterial stiffness parameters, produced by the vasodilatory effect of alcohol and the consequent decrease in SBP [81].

### 4.4. Vascular Structure and Function According to the Type of Beverage

The influence of the type of alcoholic drink on vascular parameters has also been studied in a population without persistent COVID. With regard to vascular structure, the results are incongruent. A study carried out in Innsbruck found a positive relationship between drinking beer and the size of atherosclerotic plaque (r = 0.156, *p* = 0.004) [29]. In contrast, a different study showed a positive association between c-IMT and the consumption of wine and spirits, but not beer [58]. In the present study, we found a positive relationship between c-IMT and drinking wine and beer, but not high-alcohol content beverages; this may be due to the small sample size of the third group (26 participants). Although a beneficial effect of red wine compared to other types of alcoholic drinks has been described [82], studies comparing types of alcoholic beverages do not analyze the amount of wine consumed and, therefore, include subjects with harmful levels of alcohol use that contribute to the increase in c-IMT. A recent study on arterial stiffness (determined by the arterial stiffness index) and alcohol use describes a significant association between arterial stiffness in male beer drinkers (*p* < 0.001) and female wine drinkers (*p* < 0.004) [83]. In this study (68% women), arterial stiffness (determined by CAVI) only showed a significant association with drinking wine.

### 4.5. Limitations and Strengths

This study has the following limitations: 1. As a cross-sectional study, causality cannot be established. 2. It does not have a large sample size, nor equality of participants of both sexes (predominance of women), so comparisons between sexes should be read with caution. 3. The study was carried out in a population diagnosed with persistent COVID and, therefore, its results cannot be extrapolated to other population groups. 4. Alcohol use, smoking rate and MD adherence were assessed by self-report questionnaires. 5. Non-drinking patients were not separated from ex-drinkers. The strengths of this study are as follows: 1. It is the first study to directly analyze the impact of alcohol intake on vascular parameters in patients with persistent COVID in the Caucasian population. 2. The WHO’s agreed definition of persistent COVID was used. 3. Researchers collected the variables after training and following a standardized protocol.

## 5. Conclusions

The results of this study indicate that, compared to non-drinking, heavy drinking in patients with persistent COVID is associated with higher levels of c-IMT and cf-PWV as indicators of vascular structure and arterial stiffness (β = 1.08 (95% CI: 1.01–1.17), β = 1.37 (95% CI: 1.04–1.80), respectively). However, prospective studies with larger sample sizes are needed to confirm these results and establish causality.

## Figures and Tables

**Figure 1 nutrients-17-00703-f001:**
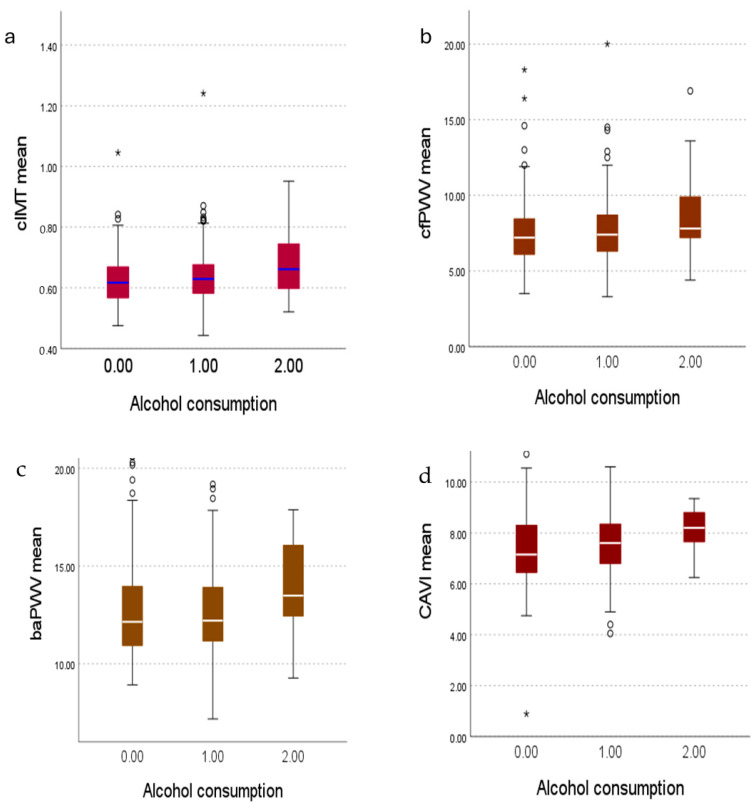
Boxplots of c-IMT, cf-PWV, ba-PWV and CAVI according to alcohol consumption. The boxes show the interquartile range, the middle line represents the median, the whiskers represent the minimum and maximum values of the variable and the points represent the values outside this range. (**a**) Data corresponding to c-IMT. (**b**) Data corresponding to cf-PWV. (**c**) Data corresponding to ba-PWV. (**d**) Data corresponding to CAVI. Abbreviations: c-IMT: intima–media thickness of common carotid; cf-PWV: carotid-femoral pulse wave velocity; ba-PWV: brachial-ankle pulse wave velocity; CAVI: cardio-ankle vascular index. * and ○ represent the values outside this range. 0.00 represents the ‘non-drinker’ group, 1.00 represents the ‘low-moderate consumption’ group, and 2.00 represents the ‘heavy consumption’ group.

**Figure 2 nutrients-17-00703-f002:**
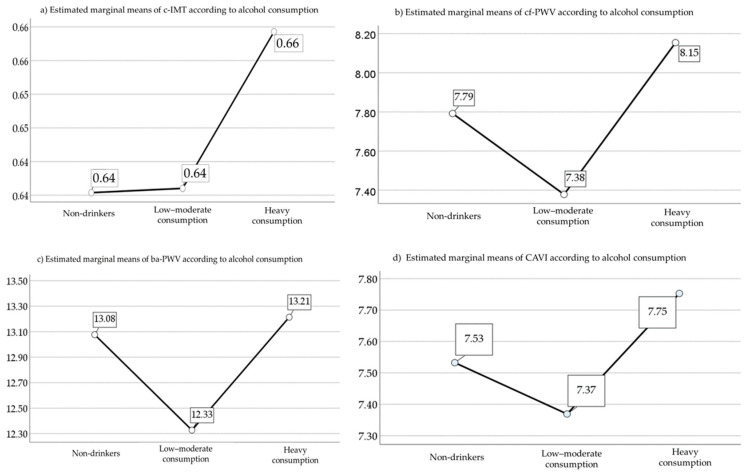
Marginal means for c-IMT, CAVI, cf-PWV and ba-PWV according to alcohol consumption adjusted for age and sex. (**a**) Data corresponding to c-IMT. (**b**) Data corresponding to cf-PWV. (**c**) Data corresponding to ba-PWV. (**d**) Data corresponding to CAVI. Abbreviations: c-IMT: intima–media thickness of common carotid; cf-PWV: carotid-femoral pulse wave velocity; ba-PWV: brachial-ankle pulse wave velocity; CAVI: cardio-ankle vascular index.

**Figure 3 nutrients-17-00703-f003:**
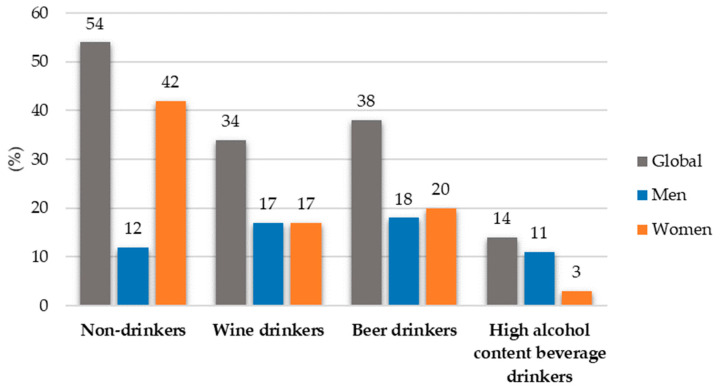
Percentage of participants according to alcohol consumption overall, men and women.

**Figure 4 nutrients-17-00703-f004:**
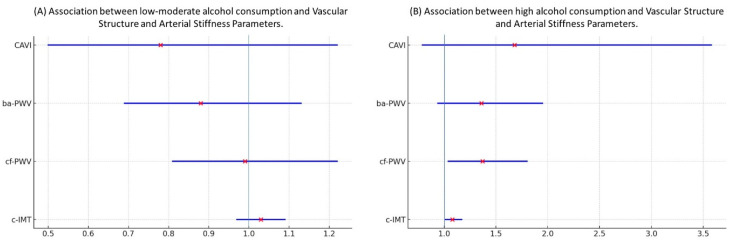
Bars show β (odds ratio) and 95% CI. Association between alcohol consumption and c-IMT, cf-PWV, ba-PWV and CAVI. (**A**) Multinomial regression analysis. Dependent variable: low-moderate alcohol consumption versus non-drinker (1 = low-moderate alcohol consumption. 0 = non-drinker). Independent variables: c-IMT, cf-PWV, ba-PWV and CAVI. Adjustment variables: age and lifestyles. (**B**) Multinomial regression analysis. Dependent variable: high alcohol consumption versus non-drinker (2 = high alcohol consumption, 0 = non-drinker). Independent variables: c-IMT, cf-PWV, ba-PWV and CAVI. Adjustment variables: age and lifestyles. c-IMT: intima–media thickness of common carotid; cf-PWV: carotid-femoral pulse wave velocity; ba-PWV: brachial-ankle pulse wave velocity; CAVI: cardio-ankle vascular index. × represents the β value along with its 95% confidence interval.

**Table 1 nutrients-17-00703-t001:** General characteristics of subjects analyzed in this study, overall and by sex.

	Overall (n = 305)		Men (n = 97)		Women (n = 208)		*p* Value
	Mean or n	SD or (%)	Mean or n	SD or (%)	Mean or n	SD or (%)	
Lifestyles							
Alcohol, (g/w)	29	53	60	76	15	27	<0.001
MD, (total score)	7.8	2.3	7.7	2.2	7.8	2.4	0.44
Years of smoking	23	12	25	12	22	11	0.11
Active smoker, n (%)	17	(5.7)	8	(8.4)	9	(4.5)	0.07
METs-min/week	5100	5000	5400	5200	5000	4900	0.22
Conventional risk factors							
Age, (years)	53	12	56	12	51	12	0.004
Time with COVID, (months)	39	9.6	39	10	38	9.4	0.99
SBP, (mmHg)	120	17	130	15	120	16	<0.001
DBP, (mmHg)	77	11	82	11	74	10	<0.001
PP, (mmHg)	43	10	48	11	41	9.3	<0.001
MAP, (mmHg)	91	12	98	11	88	12	<0.001
HR, (bpm)	70	11	72	14	70	10	0.28
Antihypertensive drugs, n (%)	79	(26)	34	(35)	45	(22)	0.01
Hypertension, n (%)	110	(36)	53	(55)	57	(28)	<0.001
FPG, (mg/dL)	88	18	94	20	85	16	<0.001
Hypoglycemic drugs, n (%)	32	(10)	18	(19)	14	(7)	<0.001
Diabetes mellitus, n (%)	37	(12)	22	(23)	15	(7)	<0.001
Total cholesterol, (mg/dL)	190	35	180	33	190	35	0.06
LDL cholesterol, (mg/dL)	110	31	110	32	110	31	0.78
HDL cholesterol, (mg/dL)	57	14	49	11	61	13	<0.001
Triglycerides, (mg/dL)	100	51	120	54	95	47	<0.001
Lipid-lowering drugs, n (%)	75	(25)	40	(42)	35	(17)	<0.001
Dyslipidemia, n (%)	172	(57)	67	(69)	105	(51)	0.003
BMI, (kg/m^2^)	28	5.6	30	4.6	27	5.8	<0.001
Obesity, n (%)	99	(33)	44	(45)	55	(26)	<0.001
Waist circumference, (cm)	94	16	100	13	89	14	<0.001
Abdominal obesity, n (%)	147	(48)	49	(51)	98	(47)	0.58
Vascular function and structure							
c-IMT, (mm)	0.64	0.09	0.68	0.12	0.62	0.07	<0.001
cf-PWV, (m/s)	7.6	2.3	8.8	2.9	7.1	1.8	<0.001
ba-PWV, (m/s)	12.8	2.4	13.6	2.4	12.4	2.3	<0.001
CAVI	7.5	1.3	7.9	1.4	7.3	1.3	<0.001

Values are means and standard deviations for continuous data, and number and proportions for categorical data. MD: Mediterranean diet; MET: metabolic equivalent of task; SBP: systolic blood pressure; DBP: diastolic blood pressure; PP: pulse pressure; MAP: mean arterial pressure; HR: heart rate; FPG: fasting plasma glucose; BMI: body mass index; c-IMT: intima–media thickness of common carotid; cf-PWV: carotid-femoral pulse wave velocity; ba-PWV: brachial-ankle pulse wave velocity; CAVI: cardio-ankle vascular index. *p* value: differences between men and women.

**Table 2 nutrients-17-00703-t002:** Characteristics of subjects analyzed by alcohol consumption.

	Non-Drinker (n = 163)		Low-Moderate Consumption (n = 121)		Heavy Consumption (n = 21)		*p* Value
	Mean	SD	Mean	SD	Mean	SD	
Lifestyles							
MD, (total score)	7.7	2.6	7.9	1.9	7.8	2.2	0.72
Years smoking ^b^	21	12	24	11	28	11	0.09
METs-min/week	4900	4800	5200	5400	6400	4700	0.42
Conventional risk factors							
Age, (years) ^a,b^	51	13	55	11	58	11	0.004
Time with COVID, (months) ^a^	38	11	40	8.0	37	8.9	0.10
SBP, (mmHg) ^b,c^	120	17	120	16	130	19	0.03
DBP, (mmHg) ^b,c^	76	12	77	10	83	13	0.03
PP, (mmHg)	44	10	42	10	47	8.5	0.14
MAP, (mmHg) ^b,c^	91	13	91	11	98	15	0.03
HR, (bpm)	71	11	70	11	70	15	0.94
FPG, (mg/dL)	88	21	87	12	94	16	0.23
Total cholesterol, (mg/dL)	190	36	190	32	180	36	0.39
LDL cholesterol, (mg/dL)	110	32	120	29	100	37	0.20
HDL cholesterol, (mg/dL)	57	14	57	14	58	12	0.89
Triglycerides, (mg/dL)	100	53	100	49	99	42	0.91
BMI, (kg/m^2^)	28	6.1	28	4.9	29	4.5	0.74
Waist circumference, (cm)	93	16	95	14	99	16	0.21
Vascular function and structure							
c-IMT, (mm) ^a,b,c^	0.62	0.08	0.65	0.10	0.69	0.11	0.003
cf-PWV, (m/s) ^b,c^	7.5	2.2	7.7	2.3	9.0	3.1	0.03
ba-PWV, (m/s) ^b,c^	12.8	2.5	12.6	2.1	14.0	2.3	0.054
CAVI ^b, c^	7.4	1.4	7.5	1.2	8.2	0.9	0.03

Values are means and standard deviations for continuous data. MD: Mediterranean diet; MET: metabolic equivalent of task; SBP: systolic blood pressure; DBP: diastolic blood pressure; PP: pulse pressure; MAP: mean arterial pressure; HR: heart rate; FPG: fasting plasma glucose; BMI: body mass index; c-IMT: intima–media thickness of common carotid; cf-PWV: carotid-femoral pulse wave velocity; ba-PWV: brachial-ankle pulse wave velocity; CAVI: cardio-ankle vascular index. *p*: statistically significant differences (*p* < 0.05). Post hoc contrasts: ^a^ Between ‘non-drinker’ and ‘low-moderate consumption’. ^b^ Between ‘non-drinker’ and ‘heavy consumption’. ^c^ Between ‘low-moderate consumption’ and ‘heavy consumption’. Rest of the groups *p* > 0.05.

**Table 3 nutrients-17-00703-t003:** Vascular structure and function according to type of drink.

	Non-Drinker (Reference) (n = 163)		Drinking Wine (n = 82)		Drinking Beer (n = 100)		High Alcohol Content Beverage (n = 26)	
	Mean	SD	Mean	SD	Mean	SD	Mean	SD
c-IMT, (mm)	0.62	0.08	0.65 ^a^	0.10	0.66 ^c^	0.11	0.66	0.08
cf-PWV, (m/s)	7.5	2.2	7.9	2.2	8.1	2.3	8.4	3.6
ba-PWV, (m/s)	12.8	2.5	12.9	2.2	13.0	2.2	13.2	2.5
CAVI	7.4	1.4	7.7 ^b^	1.1	7.7	1.1	7.7	1.2

Values are means and standard deviations for continuous data. c-IMT: intima–media thickness of common carotid; cf-PWV: carotid-femoral pulse wave velocity; ba-PWV: brachial-ankle pulse wave velocity; CAVI: cardio-ankle vascular index. ^a^ *p* = 0.02 compared with the reference category (non-drinker). ^b^ *p* = 0.03 compared with the reference category (non-drinker). ^c^ *p* = 0.01 compared with the reference category (non-drinker). Rest of the groups *p* > 0.05.

**Table 4 nutrients-17-00703-t004:** Association between alcohol use and c-IMT, cf-PWV, ba-PWV and CAVI. Multinomial regression analysis.

	Non-Drinker (n = 163)	Low-Moderate Consumption (n = 121)			Heavy Consumption (n = 21)		
	Reference	β	(95%) CI		β	(95%) CI	
c-IMT, (mm)	Reference	1.03	0.97	1.09	1.08 *	1.01	1.17
cf-PWV, (m/s)	Reference	0.99	0.81	1.22	1.37 *	1.04	1.80
ba-PWV, (m/s)	Reference	0.88	0.69	1.13	1.36	0.94	1.95
CAVI	Reference	0.78	0.50	1.22	1.68	0.79	3.58

Multinomial regression analysis using ‘alcohol consumption’ as dependent variable and ‘c-IMT’, ‘ba-PWV’, ‘cf-PWV’ and ‘CAVI’ as independent variables. Adjusted by age and lifestyles. c-IMT: intima–media thickness of common carotid; cf-PWV: carotid-femoral pulse wave velocity; ba-PWV: brachial-ankle pulse wave velocity; CAVI: cardio-ankle vascular index. * *p* < 0.05.

## Data Availability

The data supporting the findings of this study are available on ZENODO under the https://doi.org/10.5281/zenodo.14282873.

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
