# Peer review of "Relationship Between Alcohol Consumption and Vascular Structure and Arterial Stiffness in Adults Diagnosed with Persistent COVID: BioICOPER Study"

_nutrients, 2025, doi:10.3390/nu17040703_

Round 1

Reviewer 1 Report

Comments and Suggestions for Authors

The manuscript is scientific based and clearly presented.

Therefore there are 2 issues.

One is related with ethical issues. The authors should provided prof of subjects consent and endorsement of the Ethical Commission of the institution where the authors are affiliated.

Second issue is related with citation. Please highlights the similar findings not just the reference number. Is important because there are direct and indirect confirmation of own findings by different studies. Also similar values should be displayed.

Author Response

View attachments

Reviewer 2 Report

Comments and Suggestions for Authors

The study "Relationship between alcohol consumption and vascular structure and arterial stiffness in adults diagnosed with persistent COVID: BioICOPER study" by Arroyo-Romero et al. introduces an exciting argument about alcohol's interactions with the vascular system. Its aim is "...to evaluate how alcohol use is related to vascular structure and arterial stiffness in adults with persistent COVID-19."

Unfortunately, the study presents issues that make it unsuitable for publication.

Issues

Raw 37: The abbreviation of grams is preferably "g" and not "gr". If the authors prefer the last form, it must be corrected in other parts of the draft where it is correctly reported.

Raw 51: J-shaped curve. In this case, the references are not complete. Recently, new papers and statistical elaborations have doubted the existence of the J-shaped curve and interpretation (Miller et al. 2024, "Still rethinking the J‐shaped curve: A commentary on Kember et al.", ACER.).

The authors use the cases of another study built for other aims; this gives less force because retrospective studies can provide only suggestions for future studies.

The subgroups are not well defined: 

1. Non-drinker: In this category, how many men have never consumed alcohol in the past or drank and stopped drinking? This definition is essential because metabolism and brain connectivity can differ.

2. How was the composition made to the subgroups? The numbers in each group are very different and can influence the statistical results: There is a high prevalence of women (w/m, 2/1).

Figure 1: In the boxplots, many cases out of 3 S.D. should be withdrawn from calculations.

Table 3 and Fig. 3 do not help demonstrate the paper's aims and are not helpful.

Comments on the Quality of English Language

Minor revisions

Author Response

Ver archivos adjuntos

Reviewer 3 Report

Comments and Suggestions for Authors

The manuscript offers a new innovative perspective on alcohol use and whether it is associated with vascular structure and arterial stiffness in adults with persistent COVID. To prove their thesis, the authors use with great precision and skill, a large number of statistical methods and characteristics: a cross-sectional study including 305 individuals (97 men and 30

208 women) diagnosed with persistent COVID according to WHO criteria; Arterial stiffness was assessed by measuring cardio-ankle vascular index (CAVI) and brachial-ankle pulse wave velocity

(ba-PWV) with the VaSera VS-1500 device and carotid-femoral pulse wave velocity (cf-PWV) with the Sphygmocor device. Vascular structure was assessed by measuring carotid intima-media thickness (c-IMT) with a Sonosite Micromax ultrasound unit. Alcohol intake was assessed using a standardized questionnaire and quantified in g/week. The manuscript is written in standard English and has a clinical focus. The present study sheds new light/in great detail on the fact that high alcohol consumption in patients with persistent COVID is associated with higher c-IMT and cf-PWV values ​​than in non-drinkers. The presentation is well written; Figures and tables are informative, well described. Figure 2 is of insufficient resolution - suggest a new one. Figure 3 should be revised, and the deviation should be suggested. The discussion is well presented and substantiated. Limitations and future directions are well described.

Author Response

Ver archivos adjuntos

Round 2

Reviewer 2 Report

Comments and Suggestions for Authors

The manuscript is ready for publication as it currently stands.

Author Response

Comentarios y sugerencias para los autores

El manuscrito está listo para su publicación en el estado actual actual.

Respuesta del autor

Agradecemos sus comentarios y sugerencias, que nos han ayudado a mejorar el manuscrito. Nos complace saber que las modificaciones realizadas han sido satisfactorias.